

# A data integration framework for spatial interpolation of temperature observations using climate model data

Theo Economou[1], Georgia Lazoglou[1], Anna Tzyrkalli[1], Katiana Constantinidou[1] and Jos Lelieveld[1,2]

[1] Climate and Atmosphere Research Center, The Cyprus Institute, Nicosia, Cyprus
[2] Department of Atmospheric Chemistry, Max Planck Institute for Chemistry, Mainz, Germany

## ABSTRACT

Meteorological station measurements are an important source of information for understanding the weather and its association with risk, and are vital in quantifying climate change. However, such data tend to lack spatial coverage and are often plagued with flaws such as erroneous outliers and missing values. Alternative meteorological data exist in the form of climate model output that have better spatial coverage, at the expense of bias. We propose a probabilistic framework to integrate temperature measurements with climate model (reanalysis) data, in a way that allows for biases and erroneous outliers, while enabling prediction at any spatial resolution. The approach is Bayesian which facilitates uncertainty quantification and simulation based inference, as illustrated by application to two countries from the Middle East and North Africa region, an important climate change hotspot. We demonstrate the use of the model in: identifying outliers, imputing missing values, non-linear bias correction, downscaling and aggregation to any given spatial configuration.

# INTRODUCTION

Climate change is one of the most serious global issues today, and much scientific effort is invested into trend analysis and understanding the impact of weather on different aspects of human life. On average, temperature across the globe has been increasing and is projected to keep doing so under various scenarios. Temperature is therefore a key indicator of climate change, so it is important to understand its association with various risks. For instance, there are studies attempting to link extreme temperature with human mortality and morbidity (*Lubczyńska, Christophi & Lelieveld, 2015*), impact models aiming to understand the dynamics of infectious diseases as a function of temperature (amongst other things) (*Erguler et al., 2022*), research on the effects of temperature on crop yield (*Matiu, Ankerst & Menzel, 2017*; *Constantinidou et al., 2016*) and many more such examples.

The main challenge for such scientific endeavours is finding temperature data at a required spatial and temporal resolution. Temperature (and other meteorological variables)

Corresponding authors
Theo Economou,
t.economou@cyi.ac.cy
Jos Lelieveld, jos.lelieveld@mpic.de

are conventionally measured using weather stations, which typically lack spatial coverage. In epidemiological studies for instance, it might be difficult to relate mortality data at city level with temperature measurements from a single weather station at the city's airport. Another example is the need to compare gridded temperature data from a climate model with corresponding historical temperature measured at point locations (*Kostopoulou et al., 2009*) or obtained from gridded datasets (*Kotlarski et al., 2014*). Climate model output is often used to drive impact models and therefore it is of great importance that they are evaluated in order to avoid passing climate model uncertainty further to the impact model (*Constantinidou, Zittis & Hadjinicolaou, 2019*; *Stéfanon et al., 2015*). Yet another example is construction, for example of a nuclear power station at a specific spatial location, where building regulation necessitates information on extreme temperature at that exact location. Many more such examples exist, the point here being that temperature data are rarely available at the required location and spatial resolution.

*In situ* weather observations are probably the closest we have to the "ground truth"; however, such observations are often plagued with errors such as non-physical outliers and missing values, particularly for historical time series going back many decades, where data may be recorded manually and later digitized. The two most utilised alternative data sources for temperature, are (a) remote sensing (*e.g.*, satellite) and (b) reanalysis products. Both of these are gridded (*e.g.*, 10 km $\times$ 10 km spatial resolution) and thus have better spatial coverage but do not provide information over specific locations (*e.g.*, at the coordinates associated with a weather station). Moreover, both of these alternatives are biased, since they can be described as proxy rather than direct measurements. For instance, satellites measure ground temperature rather than air temperature at 2 m which is what is usually of interest to humans, and cannot measure temperature accurately on cloudy days (*Hooker, Duveiller & Cescatti, 2018*). Reanalysis data on the other hand have complete spatial and temporal coverage but they are the output of physical (climate) models and are therefore not actual measurements. Rather, they are data-informed model predictions and are possibly biased (*Rhodes, Shaffrey & Gray, 2015*).

Nevertheless, we argue that the wide availability of reanalysis products and the fact that such data respect physical mechanisms, means that they contain useful information and can be used in conjunction with *in situ* data for a robust estimate of temperature at any spatial resolution. In this article we propose a Bayesian hierarchical modelling approach to integrate *in situ* temperature measurements and reanalysis data, and demonstrate how this can be used to achieve the goal of obtaining temperature estimates at any required spatial resolution. Specifically, we look for an approach that:

1. allows for erroneous outliers in the *in situ* temperature data;
2. automatically integrates gridded reanalysis data and point *in situ* measurements (change-of-support problem);
3. has adequate flexibility to capture biases between reanalysis and station data;
4. can be used to correct and impute missing values in station observations;
5. fully quantifies the associated uncertainty.

Challenge 1 is important so that erroneous outliers do not influence the statistical properties (particularly the extremes) of any predictions. Challenge 2 is typically an issue

when combining data of different resolutions and what is required is a single robust estimate. The form of any biases between climate model output and observations are not a-priori known, so flexibility in challenge 3 is key. Observational weather data are invariably plagued with missing values and outliers, so an approach that can automatically deal with challenge 4 can increase the value of weather data. Lastly, the sometimes overlooked challenge 5 is crucial for appreciating the weight of evidence behind estimates—particularly when predicting outside the range space of the data (*e.g.*, when downscaling).

The following section provides further background and relates the work to the literature, while the "Data and related challenges" section describes the data and their associated challenges. The "Data and related challenges" section lays out the modelling framework, The "Model implementation" section describes its implementation and "Results" demonstrates application to temperature data from Cyprus and Morocco. The final section summarises and presents a discussion.

## BACKGROUND

The probabilistic modelling framework presented here aims to capture the association of temperature measurements with a gridded reanalysis data set, in a way that allows prediction of temperature at any given location and also time point within the range of the reanalysis data. The approach is therefore akin to the idea of bias correction of climate model data, but also to the idea of statistical downscaling of climate data, as well as the concept of stochastic weather generators. The distinction between these approaches is often blurred and there are many methods that can thus be classified as hybrid. In fact, the terms "bias correction" and "statistical downscaling" are used in different ways in different communities (*Maraun, 2013*). Our approach simultaneously performs bias correction and downscaling but can also be used as a stochastic weather generator, while also correcting for outliers in temperature records. The method can thus be classified as a hybrid, but nevertheless an effort is made next to place the work in the wider literature.

Bias correction methods are ubiquitous in climate science where the idea is to correct desired statistical properties of climate model data with those from observations (*Christensen et al., 2008*). Methods that assume linear bias are the simplest category and are favoured for their simplicity and computational efficiency. However, more sophisticated methods that focus on the whole probability distribution are the most promising due to their accuracy. Examples include regression approaches (*Durai & Bhradwaj, 2014*), copula-based methods (*Lazoglou, Gräler, & Anagnostopoulou, 2019*) and quantile mapping (*Maraun, 2013*). In bias correction, there is an implicit assumption that the observational data used to correct the climate output data are accurate. In general however this is not the case, where for instance flawed outliers may influence estimation of extremes thus limiting utility of bias correction (*Maity et al., 2019*)). Here, we explicitly allow for the presence of erroneous outliers in the station measurements in addition to using a flexible way of capturing the bias using penalised splines. In essence, our method can be classified as a non-linear regression approach to bias-correction, although we do so in a way that separates linear and non-linear terms, for more robust spatial extrapolation.

Statistical (as opposed to dynamical) downscaling (*Maraun & Widmann, 2018*) is a technique for increasing the spatial resolution of climate model data, say from a 10 km ×10 km spatial grid to a 1 km ×1 km grid. Conventionally this involves quantifying the relationship between climate model data at a coarse resolution with higher resolution gridded observational data (such as reanalysis *e.g.*, *Hernanz et al. (2022)*), although more recent methods (*e.g.*, *Huth, 2002*) also include *in situ* observations. A related method to downscaling is the concept of spatial interpolation/extrapolation of weather observations, on the basis of meaningful predictive information (*i.e.,* systematic local effects such as elevation and distance from the sea). For instance, *Camera et al. (2014)* have produced a gridded precipitation data set by quantifying the relationship of weather station data with topographical information. Another example is *Lompar et al. (2019)* who, like here, use ERA5 reanalysis to impute missing temperature measurements in time series. Our approach can be seen as a combination of statistical downscaling and interpolation, where the spatial interpolation is of the relationship between the temperature observations and the climate model data, while also allowing for inclusion of local covariates either additively or by extending the spatial interpolation to include additional dimensions. However, here we have the added requirement that erroneous outliers in the observations are allowed for in addition to requiring the model predictions to be put at any spatial configuration. The latter is achieved by interpreting the predictions as simulations from a random field, which can be integrated over any spatial unit (*Poole & Raftery, 2000*).

Stochastic weather generators are typically probabilistic modelling tools with which existing weather data sets can be expanded temporally and spatially. Such generators have been utilised by the water industry for instance to quantify flood/drought risk (*Dawkins et al., 2022*; *Stoner & Economou, 2020*) as well as the reinsurance sector for estimating natural hazard risk (*Youngman & Economou, 2017*). The approach presented here can be viewed as a stochastic weather generator, with the added benefit of filtering erroneous outliers so that probabilistic simulations of temperature are not unduly affected.

Lastly, the model we present can also be seen as method with which one can identify outliers in temperature records with the help of the physically constrained reanalysis data. Outlier detection is a well established field in data science (*Hawkins, 1980*; *Barnett & Lewis, 1994*; *Hodge & Austin, 2004*; *Jobe & Pokojovy, 2015*) and also more specifically in temperature modelling (*Ma, Gu & Wang, 2017*; *Sun et al., 2015*; *Li & Jung, 2021*). In this work, outliers are identified simultaneously with modelling the data and thus can be classified as a ''Type 2'' outlier-detection method (*Hodge & Austin, 2004*). The novelty of the method is the use of a discrete mixture distribution to identify outliers in conjunction with a Bayesian hierarchical model for modelling the temperature data, and penalised splines to characterise the association with the reanalysis data.

The novelty of our approach lies in the combination of the challenges it aims to tackle: bias-correction, spatial aggregation and downscaling, outlier detection/correction and stochastic simulation. In this sense, the approach is unique and not easily comparable. The mathematical complexity is kept as simple as possible in order to emphasize interpretability and out-of-sample performance, so the individual aspects of our modelling framework are

probably simpler than the state-of-the-art. For instance, bias correction here is based on non-linear regression which may not be as flexible as some copula methods.

## DATA AND RELATED CHALLENGES

One of the main motivations behind this work is the study of temperature and its impacts in the Middle East and North Africa (MENA) region, for instance in understanding the association between maximum temperature and mortality in the region. We focus on two countries in this region, namely Cyprus and Morocco, in order to assess its applicability to different geographical regions and sizes.

### In situ data

Daily measurements of maximum temperature (Tmax) at 2 m height were obtained from the Global Surface Summary of the Day (GSOD) which is derived from The Integrated Surface Hourly dataset (*GSOD, 2022*). There are 17 weather stations in Cyprus shown in Fig. 1 and 40 stations in Morocco (Fig. S5). For brevity we mostly focus on the Cyprus data, although we show some results relating to Morocco later on. Table 1 shows the temporal span of the data for each station, the elevation in meters and the proportion of missing values. Note that the stations are basically scattered around the coastline with little inland coverage particularly in terms of elevation (the middle and midwest of Cyprus where we have no data, have mountains reaching up to 2,000 m while the highest station in our data set is 217 m).

Figure 2 shows the time series of Tmax for the 17 stations, where missing values are clearly an issue both in terms of temporal span, but also in-between the sampling periods. The same table is provided for Morocco in the supplementary material. If we look at specific time snaps, for instance at station 16 between 1978 and 1982 shown in Fig. 3A, we can see that spurious outliers are apparent. Such outliers also appear in the Morocco time series (Fig. S6).

### Reanalysis data

The data utilised here is the European Centre for Medium-Range Weather Forecasts (ECMWF) Reanalysis v5 or ERA5-Land (*Muñoz Sabater, 2021*; *Muñoz Sabater et al., 2021*). The ERA5-Land dataset has gridded hourly temperature at 2m height available from 1st January 1950 to 31st December 2020 at a 0.1° latitude × 0.1° longitude resolution (approximately 11 ×11 km). The reanalysis data set combines model data with observations from across the world and it is produced using 4D-Var data assimilation and model forecasts in CY41R2 of the ECMWF Integrated Forecast System (IFS). The maximum of the hourly temperature values in a given day was used as an estimate of the daily Tmax from ERA5-Land.

The ERA5-Land grid configuration over Cyprus is given in Fig. 1 which shows the mean daily Tmax in each grid cell over 1950–2020. Due to the coarseness of the grid, not all stations correspond to an ERA5-Land cell (*e.g.*, stations 11 and 12). To match each station with an appropriate ERA5-Land grid cell, we associate a distance-weighted average of the ERA5-Land Tmax for the 10 nearest-neighbour cells of each station. For the remainder

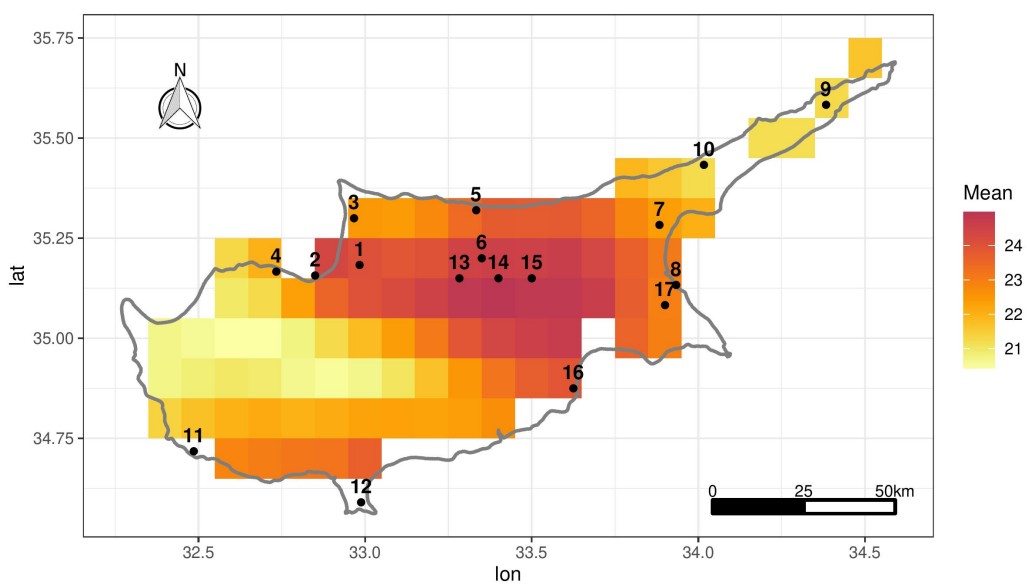

**Figure 1** **Location of the 17 stations in Cyprus in black.** The grid shows ERA-Land grid cells over Cyprus illustrating mean daily Tmax over the period 1950–2020 in each grid cell.

**Table 1** **Cyprus the weather station information.** The elevation is in meters and the fifth column is the proportion of missing values for each station.

| Station Number | Station Name | Elevation | Temporal span | Proportion missing | Proportion outliers |
|---|---|---|---|---|---|
| 1 | GUZELYURT | 52.000 | 23/03/05–31/12/20 | 0.090 | 0.000 |
| 2 | LEFKE | 129.000 | 21/07/08–06/12/20 | 0.010 | 0.001 |
| 3 | AKDENIZ | 89.000 | 21/07/08–31/12/20 | 0.020 | 0.000 |
| 4 | YESILIRMAK | 20.000 | 01/07/09–31/12/20 | 0.030 | 0.000 |
| 5 | GIRNE | 10.000 | 23/03/05–31/12/20 | 0.050 | 0.000 |
| 6 | LEFKOSA | 131.000 | 04/07/05–31/12/20 | 0.040 | 0.001 |
| 7 | ISKELE | 39.000 | 21/07/08–31/12/20 | 0.020 | 0.003 |
| 8 | GAZIMAGUSA | 0.000 | 23/03/05–31/12/20 | 0.080 | 0.002 |
| 9 | DIPKARPAZ | 136.000 | 01/07/09–31/12/20 | 0.090 | 0.005 |
| 10 | YENIERENKOY | 123.000 | 21/07/08–31/12/20 | 0.030 | 0.001 |
| 11 | PAFOS INTERNATIONAL | 12.490 | 01/01/50–31/12/20 | 0.110 | 0.001 |
| 12 | AKROTIRI | 23.160 | 03/01/60–31/12/20 | 0.080 | 0.001 |
| 13 | NICOSIA AIRFIELD | 216.700 | 01/01/50–19/07/74 | 0.170 | 0.002 |
| 14 | NICOSIA ATHALASSA | 161.000 | 01/01/90–03/11/20 | 0.460 | 0.001 |
| 15 | ERCAN | 91.000 | 24/09/93–31/12/20 | 0.150 | 0.010 |
| 16 | LARNACA | 2.430 | 14/01/77–31/12/20 | 0.000 | 0.001 |
| 17 | AYIOS NICOLAOS | 37.000 | 15/12/54–29/09/77 | 0.220 | 0.002 |

of the article, including the exploratory analysis that follows, the ERA5-Land values that are used are actually 10-cell weighted averages. This also alleviates the choice of a single "representative" grid cell for each station.

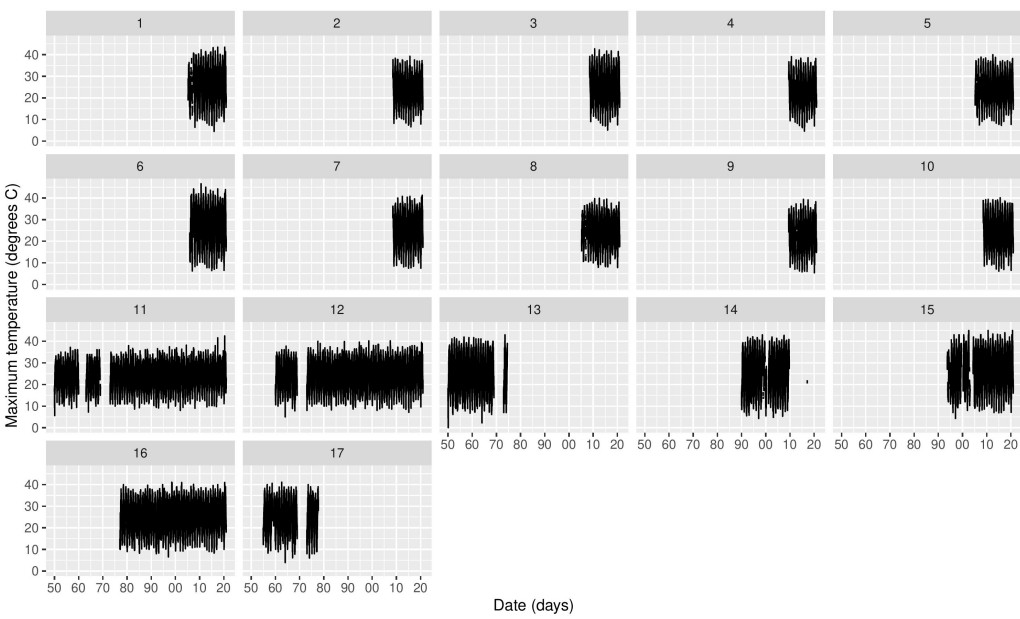

**Figure 2** **Time series of daily Tmax for each station in Cyprus.** The *x*-axis tick marks are in decades.

Figure 4A shows scatterplots of Tmax from four selected Cyprus stations plotted against the corresponding ERA5-Land Tmax. The plots were selected as a summary of the overall picture: a strong and approximately linear relationship. However, the slope of the apparent linear relationship varies, while some stations like 2 (Lefke) and 11 (Pafos) also exhibit non-linearity. Exploratory analysis (not shown) indicates that such non-linearities do not appear to be systematic *e.g.*, they are not a function of coordinates or elevation or proximity to the coast. On the other hand, approximating the relationship with a linear (regression) fit indicates that stations in close proximity to each other exhibit a similar structure, as shown in Fig. S1 of the online supplementary material (*e.g.*, stations 1–4 and also 6, 13–15). A qualitatively similar picture is also seen across the Morocco stations.

## MODELLING FRAMEWORK

First, let $y_{s_j,t}$ denote Tmax measured by a weather station $j = 1, \ldots, J$ on day $t$ and spatial location $s_j$ (defined by the spatial coordinates of station $j$). Also let $x_{s_j,t}$ denote the corresponding ERA5-Land weighted average of Tmax. To allow for erroneous outliers we formulate a discrete mixture distribution which we define conditionally on a latent Bernoulli variable $z_{j,t}$ so that:

$$y_{s_j,t} | z_{j,t} = 1 \sim N(\mu_{s_j,t}, \sigma_j^2) \tag{1}$$

$$y_{s_j,t} | z_{j,t} = 0 \sim Unif(U_{min}, U_{max}) \tag{2}$$

$$z_{j,t} \sim Bern(\pi_j) \tag{3}$$
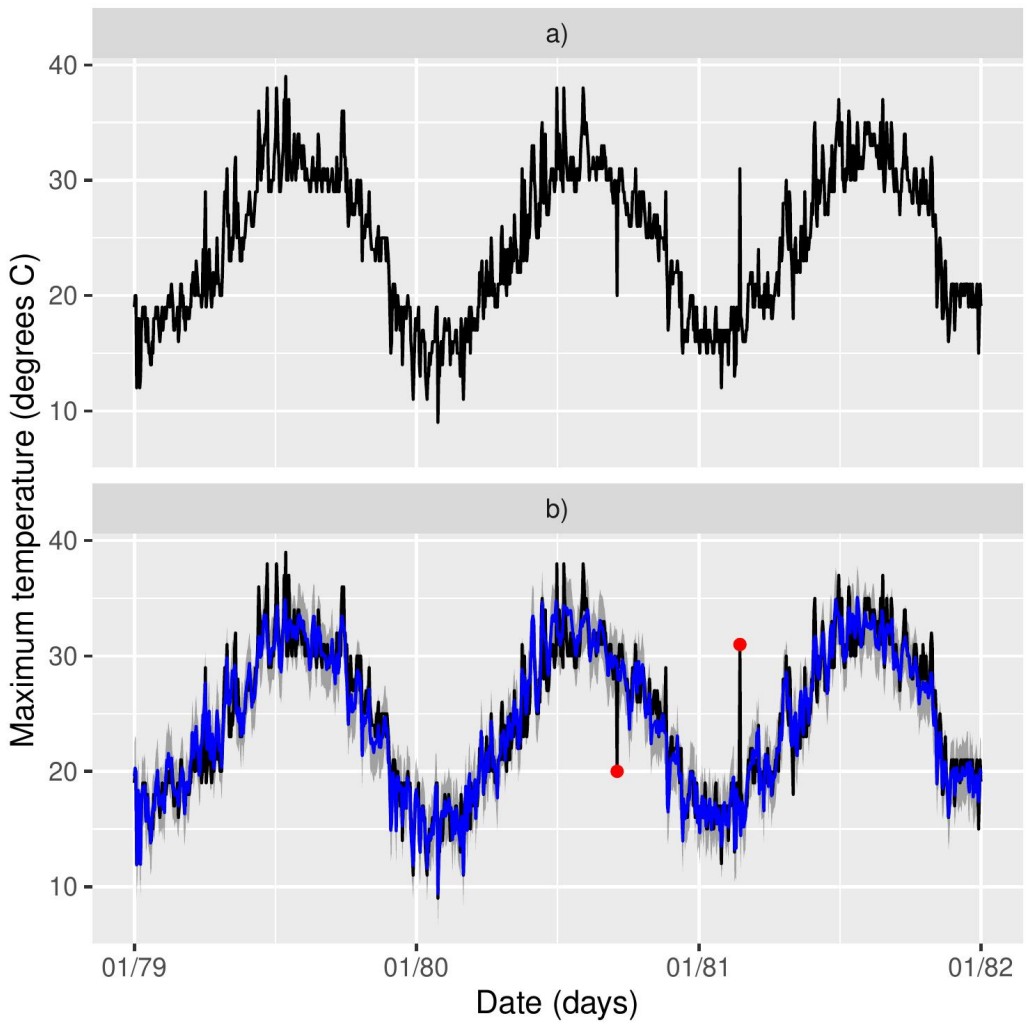

**Figure 3** **(A–B) Station 16 (Larnaca) timeseries for the period 1978–1982 and model predictions in blue.** A suspicious outlier seems present towards the end of 1980 and another one at the start of 1981.

so that $(1-\pi_j)$ is the proportion of outliers in station $j$, and $t = 1,\ldots,n_j$ where $n_j$ is the number of data points in station $j$. Conditional on $z_{j,t}$, temperature is thus described by a Normal distribution in a way that each station is allowed its own variance $\sigma_j^2$ (an assumption supported by Fig. 2 where temperature variability is different across stations). The outliers are conditionally modelled by a Uniform distribution since we have no knowledge of the outlier-generating mechanism, and here we set $U_{min} = -80°C$ and $U_{max} = 80°C$ as the boundaries. These values exceed ones that are physically plausible and also ones in both the Cyprus and Morocco data. The idea is that if any given data point $y_{s_j,t}$ is too extreme with respect to Eq. (1), then it is captured by Eq. (2), both of which in turn inform estimation of Eq. (3). In trial runs we found little sensitivity (in outlier-detection) to the choice of the bounds, as long as these are large enough with respect to the range space of the data.

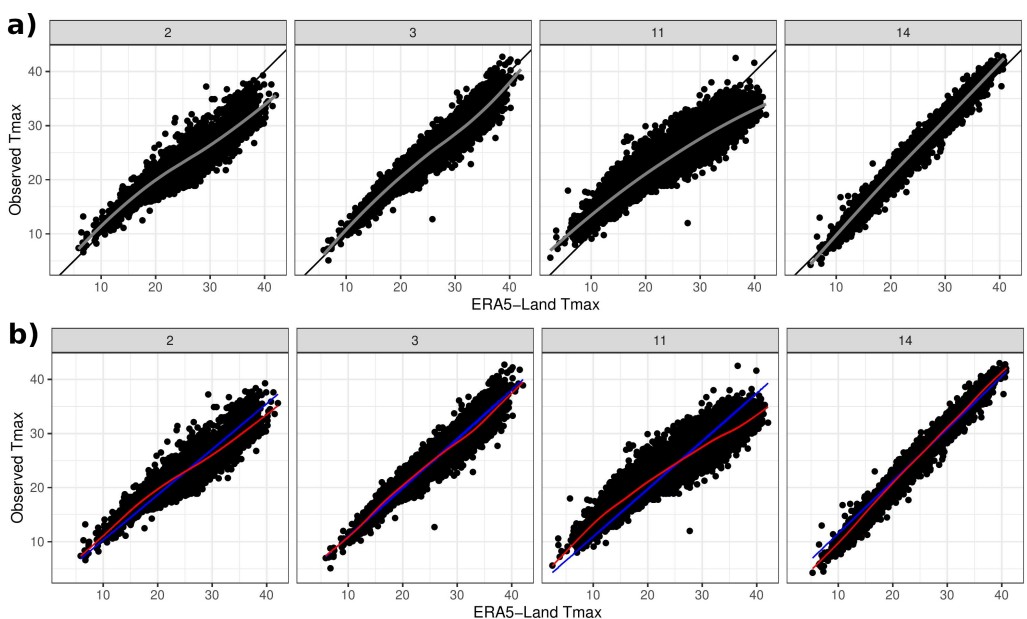

**Figure 4** (A) Observed Tmax at four weather stations *vs* Tmax from ERA5-Land. Black lines depict the 45° line and grey lines relate to a LOESS fit. (B) Model estimates of the linear term $f(s_j) + g(s_j)x_{s_j,t}$ in blue and the linear plus non-linear term $\mu_{s_j,t}$ in red.

The mean $\mu_{s_j,t}$ is then modelled as a function of $x_{s_j,t}$ viz:

$$\mu_{s_j,t} = \alpha_0 + f(s_j) + g(s_j)x_{s_j,t} + h_j(x_{s_j,t}) \tag{4}$$

where $f(\cdot), g(\cdot)$ and $h(\cdot)$ are smooth functions. The first three terms describe a spatially varying linear relationship, where both intercept $\alpha_0 + f(s)$ and slope $g(s)$ vary smoothly as functions of the coordinates. This is designed in order to reflect the findings of the exploratory analyses, *i.e.,* the apparent linear relationship with ERA5-Land being similar in neighbouring locations. The last term $h_j$ is a station-specific function of the covariate $x_{s_j,t}$ and its purpose is to capture non-linearity in the relationship. It is not a function of space, rather it can be thought as a 'random effect' term aimed at capturing station-specific behaviour.

The particular formulation of $\mu_{s_j,t}$ is based on the requirement for spatial interpolation and extrapolation (downscaling). Given the spatial sparsity of weather stations compared to the domain size, it is more robust to spatially downscale the linear part of the relationship. The non-linear part is constructed in way that it can be integrated out when predicting at unseen locations as shown later in "Results". The following subsection describes how the smooth functions are constructed and then the rest of the model components are defined.

## Bayesian penalised splines

A smooth function of some covariate $x_i$ say, can be constructed using regression splines *via* a linear combination of basis function (*Wood, 2017*) e.g.,

$$f(x_i) = \sum_{k=1}^{K} \beta_k b_k(x_i) = X_i \boldsymbol{\beta} \tag{5}$$

where $\boldsymbol{\beta} = \{\beta_k\}$ are unknown coefficients ($k = 1$ conventionally aliased to an intercept) and $b_k(\cdot)$ are basis functions. Matrix $X_i = \{b_k(x_i)\}$ with dimension $n \times K$ ($n$ being the number of data points) is the model matrix. The value of $K$ (conventionally the number of knots) determines the flexibility of $f(\cdot)$. Regression models involving such smooth functions can be estimated using penalised likelihood, where the penalty is in restricting the amount of flexibility in $f(\cdot)$ in order to avoid overfitting (*Wood, 2011*). Specifically, the log-likelihood to be maximised can be written as

$$\ell(\boldsymbol{\beta}, \theta; \boldsymbol{y}) - \lambda_\beta \boldsymbol{\beta}' S_\beta \boldsymbol{\beta} \tag{6}$$

where $\ell(\cdot)$ is the log-likelihood, $\theta$ are other model parameters and $\lambda_\beta$ is a penalty parameter. Moreover, $S_\beta$ is a penalty matrix that relates to a quadratic penalty on $\boldsymbol{\beta}$, and is basically a function of the particular basis functions chosen, as well as any constraints on the function. For instance, $f(s)$ in equation Eq. (4) is centered on zero to identify the overall mean intercept $\alpha_0$. The second term in Eq. (6) penalises the flexibility (wiggliness) of $f(\cdot)$ so the penalty increases with $\lambda_\beta$.

From a Bayesian perspective, the smoothness of $f(\cdot)$ can be viewed as a constraint on the values of $\boldsymbol{\beta}$, which one can express in the form of an appropriate prior distribution (*Wood, 2016*; *Wood, 2017*). Specifically,

$$\boldsymbol{\beta} \sim N\left(0, \boldsymbol{\Omega}_\beta^{-1} = S_\beta^{-1}/\lambda_\beta\right). \tag{7}$$

This prior is improper since the precision matrix $\boldsymbol{\Omega}_\beta = \lambda_\beta S_\beta$ is usually rank deficient (*Wood, 2016*). Instead, we can use the precision matrix $\boldsymbol{\Omega}_\beta = \lambda_\beta^{(0)} S_\beta^{(0)} + \lambda_\beta S_\beta$ where $S_\beta^{(0)}$ relates to the penalty of the null space of $f(\cdot)$ and $\lambda_\beta^{(0)}$ is the corresponding penalty parameter. This can be interpreted as separating the penalty matrix into penalised components $S_\beta$ (*e.g.*, wiggly behaviour) and unpenalised components $S_\beta^{(0)}$ (*e.g.*, intercept and linear terms) (*Pedersen et al., 2019*; *Wood, Scheipl & Faraway, 2013*). This decomposition is exploited in defining function $h_j(x)$ in Eq. (4), by not including the null space components. This way, the sum $g(s)x + h_j(x)$ in Eq. (4) is a non-linear smooth function of $x$, decomposed into a linear part plus a non-linear "deviation". Such penalty matrices and corresponding model matrices are readily provided by the R function `jagam` (*Wood, 2016*) from the R package `mgcv`.

## Specification of the mean

Returning to the smooth functions $f(\cdot), g(\cdot)$ and $h(\cdot)$ in Eq. (4), we choose thin-plate splines (*Wood, 2017*) as the basis functions for all of them. This particular basis can be used to define smooth functions of more that one variable whilst keeping the number of knots

small. The slope term (dropping the station subscript $j$ for clarity) is defined as:

$$g(s) = g(lon_s, lat_s) = X^{(g)}\beta \tag{8}$$

$$\beta = (\beta_1, \ldots, \beta_{N_\beta}) \sim N(\mathbf{0}, \Omega_\beta^{-1}) \tag{9}$$

$$\Omega_\beta = \sum_{i=1}^{2} \lambda_\beta^{(i)} S_\beta^{(i)}. \tag{10}$$

where $X^{(g)}$ is the associated model matrix corresponding to the thin-plate splines of the coordinates. There are two penalty parameters, one for the null space and one for the wiggly part of $g(\cdot)$. We set $N_\beta = J - 1$, i.e., the total number of stations minus one (the maximum allowed given we only have $J$ spatial locations), in order to a-priori give maximum flexibility should it be required.

The intercept term is defined in exactly the same way, except that we incorporate the overall mean $\alpha_0$ in the vector of coefficients:

$$\alpha = (\alpha_0, \alpha_1, \ldots, \alpha_{N_\alpha}) \tag{11}$$

$$f(s) = f(lon_s, lat_s) = X^{(f)}\alpha^{[-1]} \tag{12}$$

$$\alpha^{[-1]} \sim N(\mathbf{0}, \Omega_\alpha^{-1}) \tag{13}$$

$$\Omega_\alpha = \sum_{i=1}^{2} \lambda_\alpha^{(i)} S_\alpha^{(i)} \tag{14}$$

$$\alpha_0 \sim N(\mu_{\alpha_0}, \sigma_{\alpha_0}^2), \tag{15}$$

where the $[-1]$ superscript denotes a vector without its first element and $N_\alpha = J - 1$ as before. We set $\mu_{\alpha_0} = 0$ and $\sigma_\alpha^2 = 25$ to express a no prior beliefs about the value of the intercept but to also not allow it physically implausible values. (Recall that $\alpha_0$ is the overall intercept when the value of ERA5-Land is zero, so it is reasonable to set the prior mean to zero given Fig. 4.) Note also that the full prior for $\alpha$ is

$$\alpha \sim N\left(\mu_{\alpha_0} = \begin{pmatrix} \mu_{\alpha_0} \\ \mathbf{0} \end{pmatrix}, \Omega_{\alpha_f}^{-1} = \begin{pmatrix} 1/\sigma_{\alpha_0}^2 & \mathbf{0} \\ \mathbf{0} & \Omega_\alpha \end{pmatrix}^{-1}\right). \tag{16}$$

Finally, the non-linear effect of the ERA5-Land covariate is defined as

$$h_j(x) = X_j^{(h)}\gamma_j \tag{17}$$

 

$$\boldsymbol{\gamma}_j = (\gamma_{1,j}, \ldots, \gamma_{N_\gamma, j}) \sim N(\boldsymbol{0}, \boldsymbol{\Omega}_\gamma^{-1}) \tag{18}$$

$$\Omega_\gamma = \lambda_\gamma S_\gamma. \tag{19}$$

where $X_j^{(h)}$ is the model matrix of ERA5-Land values corresponding to station $j$ and $N_\gamma = 8$, since exploratory analysis indicates that the non-linearity is not severe. There is only one penalty parameter as the function does not incorporate a linear term. Note also that while each station $j$ has its own function $h_j(x)$, they share a common penalty parameter $\lambda_\gamma$ in order to pool information across the stations in this respect. Interpreting the function $h_j(x)$ as a "random effect", is also desirable in terms of being able to integrate it out when predicting at unseen locations.

### Outlier mechanism

The outliers are modelled by a Uniform distribution, where station-specific parameter $\pi_j$ determines the proportion of non-outliers. We model $\pi_j$ hierarchically to further pool information across stations in this respect. Specifically,

$$\pi_j \sim Beta(\alpha_\pi, \beta_\pi), \tag{20}$$

where we chose $\alpha_\pi = 5$ and $\beta_\pi = 2$ so that the mean and standard deviation of $\pi_j$ are 0.71 and 0.16. This way, more weight is given to values closer to 1, on the belief that most of the data points are not outliers.

### Conditional variance

Each station is given its own conditional variance, to allow for station-specific variability about the mean (see Fig. 2). This is also done hierarchically:

$$\sigma_j^2 \sim InvGamma(\alpha_\sigma, \beta_\sigma). \tag{21}$$

so that again information is pooled across stations and one can integrate this parameter out when predicting in unseen locations. This prior is chosen to enable conditional conjugacy of $\sigma_j^2$ with the Gaussian likelihood. The hyperparameter $\alpha_\sigma$ is fixed to the value of 2, so that $\beta_\sigma$ controls both the mean and variance of this distribution. Hyperparameter $\beta_\sigma$ is given an $Exp(0.1)$ prior with mean 10 and variance 100, to obtain a reasonably flat prior. Given $\alpha_\sigma$ and $\sigma_j^2$, $\beta_\sigma$ is conjugate Gamma.

### Penalty parameters

Lastly, for all penalty parameters $\lambda_\alpha^{(i)}$, $\lambda_\beta^{(i)}$ and $\lambda_\gamma$ the half-Cauchy distribution (*Gelman et al., 2013*) with scale parameter 20 was chosen. Since larger values of $\lambda$ imply more penalisation and therefore more smoothing, this heavy tailed prior was chosen to allow a wide range of values and therefore a wide range of wiggly behaviour of the smooth functions.

## MODEL IMPLEMENTATION

The model is implemented using MCMC and in particular Gibbs sampling for all model unknowns except for the penalty parameters. Some of the prior choices were made specifically to enable conditional conjugacy to be exploited for computational efficiency.

### Sampling the outliers

Let $\Theta$ denote the list of all model unknowns and $y$ denote the vector of all data points. The full conditional for $z_{j,t}$ in Eq. (3) is

$$p(z_{j,t} = 1 | \Theta, y) \propto \frac{\pi_j}{\sqrt{2\pi}\sigma_j} \exp\left\{ 2\sigma_j^{-2}(y_{s_j,t} - \mu_{s_j,t})^2 \right\} \tag{22}$$

$$p(z_{j,t} = 0 | \Theta, y) \propto \frac{1 - \pi_j}{U_{max} - U_{min}}, \tag{23}$$

where we can reconcile proportionality by dividing Eqs. (22) and (23) by their sum.

### Sampling the outlier proportions

Conditional on samples of $z_{j,t} | \Theta, y$, the proportions $\pi_j$ are sampled from their full conditional

$$\pi_j | z, y, \Theta \sim Beta\left( \alpha_\pi + \sum_t z_{j,t}, \beta_\pi + n_j - \sum_t z_{j,t} \right) \tag{24}$$

using the fact that the Beta distribution is the conjugate prior of the Bernoulli proportion parameter (*Fink, 1997*).

### Sampling the conditional variance

For the remainder of this section, we exploit conditional conjugacy when the likelihood is Gaussian, and therefore all results are presented conditionally on $z_{j,t} = 1$. Therefore, only data points corresponding to $z_{j,t} = 1$ contribute to the estimation of the non-outlier part of the model *i.e.,* (1). As such, when vector $y$ and model matrices such as $X^{(f)}$ are used, they exclude indices or rows that correspond to $z_{j,t} = 0$.

Given $z_{j,t}$, the variances $\sigma_j^2$ can be sampled from their full conditional (*Fink, 1997*):

$$\sigma_j^2 | z_j, y, \Theta \sim InvGamma\left( \alpha_\sigma + n_j/2, \beta_\sigma + \sum_t (y_{j,t} - \mu_{j,t})^2 / 2 \right) \tag{25}$$

where $n_j$ and the sum exclude any data points flagged as outliers by $z_{j,t} | \Theta, y$.

### Sampling the spline coefficients

For the coefficients, we use the following result. If $\theta \sim N(Q^{-1}b, Q^{-1})$ then $\theta \sim N_C(b, Q)$ is the canonical parameterisation of the multivariate Normal. Now suppose that $\theta \sim N_C(b, Q)$ (*i.e.,* the prior) and also that $y | \theta \sim N(\theta, P^{-1})$ (*i.e.,* the conditional likelihood). Then,

$$\theta | y \sim N_C(b + Py, Q + P). \tag{26}$$

gives the full conditional for $\theta$ (Lemma 2.2 from *Rue & Held (2005)*).

We begin with the coefficients $\boldsymbol{\alpha}$ of the intercept term. Let $W|z = y - X^{(g)}\boldsymbol{\beta} - X^{(h)}\boldsymbol{\gamma}$ where $\boldsymbol{\gamma} = (\boldsymbol{\gamma}_1, \ldots, \boldsymbol{\gamma}_J)$ and $X^{(h)} = (X_1^{(h)}, \ldots, X_J^{(h)})$. Since $y_{s_j,t}|z_{j,t} = 1$ is Gaussian,

$$W|z, \Theta \sim N\left(X^{(f)}\boldsymbol{\alpha}, \Sigma\right) \tag{27}$$

where $\Sigma = \mathrm{diag}(\sigma_1^2, \ldots, \sigma_J^2)$ is a diagonal matrix such that each $\sigma_j^2$ is repeated $n_j$ times. Pre-multiplying Eq. (27) by $(X'^{(f)}X^{(f)})^{-1}X'^{(f)}$ gives

$$(X'^{(f)}X^{(f)})^{-1}X'^{(f)}W|\boldsymbol{\alpha} \sim N\left(\boldsymbol{\alpha}, \Sigma(X'^{(f)}X^{(f)})^{-1}\right). \tag{28}$$

The prior on $\boldsymbol{\alpha}$ is also Normal (see Eq. (16)) so using equation (26) its full conditional is

$$\boldsymbol{\alpha}|W, z, \Theta \sim N_C\left(\Omega_{\alpha_f}\boldsymbol{\mu}_{\alpha_0} + X'^{(f)}\Sigma^{-1}W, X'^{(f)}\Sigma^{-1}X^{(f)} + \Omega_{\alpha_f}\right). \tag{29}$$

In the same way, we can sample the slope term coefficients $\boldsymbol{\beta}$. Let $A|z = y - X^{(f)}\boldsymbol{\alpha} - X^{(h)}\boldsymbol{\gamma}$. Then,

$$A|z, \Theta \sim N\left(X^{(g)}\boldsymbol{\beta}, \Sigma\right) \tag{30}$$

$$\implies (X'^{(g)}X^{(g)})^{-1}X'^{(g)}A|\boldsymbol{\beta} \sim N\left(\boldsymbol{\beta}, \Sigma(X'^{(g)}X^{(g)})^{-1}\right). \tag{31}$$

so that the full conditional is

$$\boldsymbol{\beta}|A, z, \Theta \sim N_C\left(X'^{(g)}\Sigma^{-1}A, X'^{(g)}\Sigma^{-1}X^{(g)} + \Omega_\beta\right). \tag{32}$$

Finally, the station specific coefficients $\boldsymbol{\gamma}_j$ are sampled similarly for each station $j$. Let $B_j|z_j = y_j - X_j^{(f)}\boldsymbol{\alpha} - X_j^{(g)}\boldsymbol{\beta}$, where $y_j$ are the response values in station $j$ and $X_j^{(f)}$ and $X_j^{(g)}$ are the row-subsets corresponding to station $j$ of $X^{(f)}$ and $X^{(g)}$ respectively. As before,

$$B_j|z_j, \Theta \sim N\left(X_j^{(h)}\boldsymbol{\gamma}_j, \Sigma_j\right) \tag{33}$$

where $\Sigma_j = \mathrm{diag}(\sigma_j^2)$. Mirroring Eq. (29), equations Eqs. (26) and (18) give:

$$\boldsymbol{\gamma}_j|B_j, z_j, \Theta \sim N_C\left(\left(1/\sigma_j^2\right)X_j'^{(h)}B_j, \left(1/\sigma_j^2\right)X_j'^{(h)}X_j^{(h)} + \Omega_\gamma\right). \tag{34}$$

## Sampling the hyperparameters

The hyperparameter $\beta_\sigma$ of the conditional variance $\sigma_j^2$ is sampled from its full conditional:

$$\beta_\sigma|\boldsymbol{\sigma}^2, \alpha_\sigma \sim Gamma\left(c + J\alpha_\sigma, d + \sum_{j=1}^{J} 1/\sigma_j^2\right) \tag{35}$$

where recall that $J$ is the number of stations. All penalty parameters($\lambda_\alpha^{(i)}$, $\lambda_\beta^{(i)}$ and $\lambda_\gamma$) are sampled using random walk Metropolis–Hastings (*Gelman et al., 2013*), with acceptance rate tuned to be in the region $[0.2, 0.5]$.

## RESULTS

All code was written in R (*R Core Team, 2022*) to sample from the full conditional distributions derived in the previous section. The Cyprus data consists of 135,471 data points, so in total 135,681 unknowns were sampled (of course 135,471 of those are the $z_{j,t}$ in equations Eqs. (22)–(23). The code takes less than 2 hours to sample 50 K samples on an Intel i9-11900F processor.

All presented results are based on running three chains for 100 K iterations after a 50 K burn-in. After thinning (to reduce autocorrelation), six K samples were obtained for each model unknown. Convergence was assessed by looking at trace plots (*e.g.*, of the deviance shown in Fig. S2 of the online supplementary material), and by computing the multivariate potential scale reduction factor (*Gelman et al., 2013*), which was 1.07 indicating acceptable convergence.

### Outliers

The first step of the analysis is to identify erroneous outliers using the model. It is important to do this before checking the model, since here the outliers are modelled by a Uniform distribution. This reflects the fact that there is no knowledge about the outlier-generating mechanism, but it also means that prediction of individual outliers will be poor (beyond their inclusion within a prediction interval). To identify outliers, we use the posterior distribution $p(z_{j,t}|\boldsymbol{y})$. MCMC samples of $z_{j,t}|\boldsymbol{y}$ are used to compute the probability of a non-outlier *i.e.*, $p(z_{j,t}=1|\boldsymbol{y})$, and here any data point $y_{j,t}$ for which $p(z_{j,t}=1|\boldsymbol{y}) < 0.5$ is identified as an outlier. More strict choices than 0.5 are possible of course, such as only considering points as outliers if $1-p(z_{j,t}=1|\boldsymbol{y}) > 0.9$.

Figure 3B shows the outliers identified for the station 16 in red, illustrating that at least intuitively the model is identifying the correct points as outliers. (A similar plot is given in Fig. S6 for a station in Morocco.) The last column in Table 1 shows the posterior mean of $(1-\pi_j)$ *i.e.*, an estimate of the proportion of outliers in each station. The proportion of outliers is overall quite small (and similar to Morocco, see Table S1) but it varies across stations.

A basic sensitivity analysis was conducted to assess the ability of the model to capture outliers. Specifically, 500 randomly chosen data points were artificially set as outliers (but not ones that were identified as such by the model). These outliers were produced by adding/subtracting (with probability 0.5) a random sample from a $Unif(M-5, M+5)$ distribution, where $M = max(|y_{s_j,t} - mean(y_{s_j,t})|)$. Here, $M = 25\,°C$. This choice ensured that the fictitious outliers are not too "obvious", but rather close to what may be considered an extreme. In 10 trial runs, all 500 of these were correctly identified each time, providing confidence to the outlier-identification mechanism.

### Model checking

To assess the performance of the model, we used posterior predictive model checking (*Gelman et al., 2013*). This involves obtaining samples from the posterior predictive distribution (PPD) $p(\tilde{y}_{s_j,t}|\boldsymbol{y},\boldsymbol{z})$ of the response value $\tilde{y}_{s_j,t}$ at any station $j$ and day $t$. Conditioning on $\boldsymbol{z}$ implies that the predictions are only for data points not identified
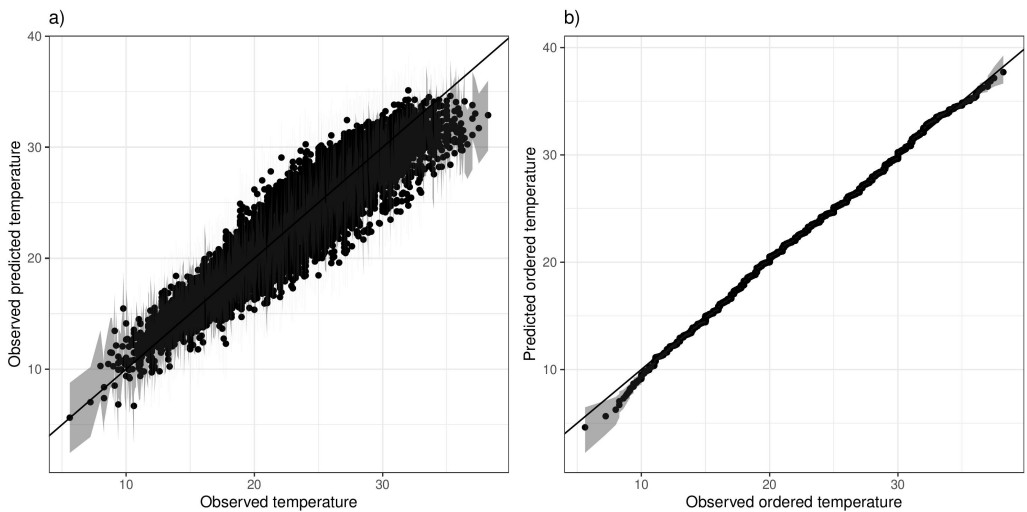

**Figure 5** (A) Predicted *vs* observed Tmax values for station 11 (Pafos). (B) Ranked Tmax values for the same station.

as outliers. The mean of samples from the PPD was used as the point estimate in Fig. 3, whereas the sample 2.5% and 97.5% quantiles were used to construct the associated prediction intervals.

We check the model both in terms of predicting the individual Tmax values, but also in terms of the overall distribution. Predictions were first compared with observations (for any non-outliers), and Fig. 5A shows this for a specific station, indicating a good fit with the exception of some under-estimation of the upper extremes. To assess whether the overall distribution is captured appropriately, we compare order statistics. Observations are sorted from smallest to largest, and compared with the corresponding ranked predictions in a plot that can be interpreted as a Q-Q plot. Figure 5B shows this for the same selected station, indicating an overall good fit albeit with some slight under-estimation of the extreme lower tail. The station in Fig. 5 was specifically chosen as the one with the least optimal model fit, while corresponding plots for the remainder of the stations are given in Figs. S3 and S4 respectively. On the whole the model fits quite well, although for some stations the predictions slightly underestimate the very high extremes. The overall distribution is captured well across stations, with no systematic discrepancies. A qualitatively similar picture is apparent for the Morocco stations (Figs. S7 and S8), for both the individual predictions but also the overall distribution.

Since we use the model for extrapolation to unobserved locations, it is also important to check the out-of-sample performance. For this reason we perform K-fold cross validation, where each station is left out in turn and then its values predicted. Figure 6 shows the associated predictions against observations for all 17 stations. Model performance is very good for all stations, even stations 11 and 12 that are isolated compared to the rest. As a summary, we define the posterior mean of the PPD for each data point as a point estimate, and in Table 2 we compare (a) the overall mean, (b) the 5% and (c) the 95% quantile of

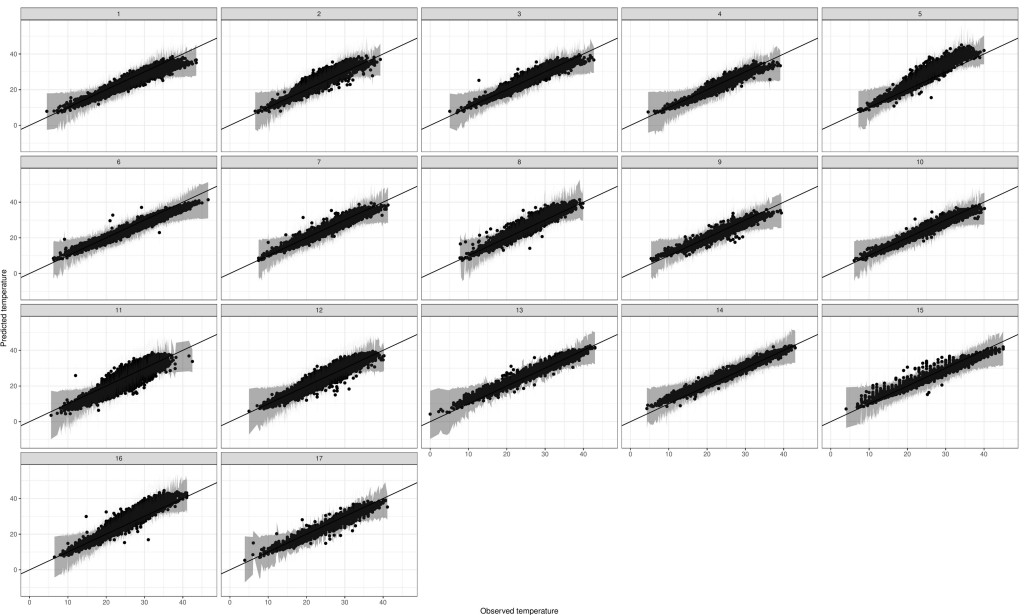

**Figure 6** Predicted *vs* observed Tmax values for the leave-one-station-out experiment.

daily Tmax for each station, against the corresponding point estimates. The table indicates high accuracy in the predictions, with deviations generally smaller than 2 °C for the mean and lower quantile. However, deviations increase for the upper quantile, reflecting the in-sample results where high extremes are slightly underestimated.

### Relationship with the reanalysis data

Figure 4B shows the estimates (posterior mean) of both the mean relationship Eq. (4) and also just the linear part $f(s_j) + g(s_j)x_{s_j,t}$ for the four chosen stations. The non-linear behaviour qualitatively matches the exploratory analysis in Fig. 4A. The model can also be used to impute missing values over the period 1950–2020 and Fig. 7 shows Tmax values for a particular station in the period 1955–1975. This station is missing values in 1960–1963 and also 1969–1973 so the model was used to impute these, along with quantifying the associated uncertainty.

### Spatial extrapolation and aggregation

One of the aims of the framework is to allow for spatial extrapolation and aggregation to various spatial configurations (*e.g.*, grids). To predict from the model at an unknown location, we must first integrate out the station-specific terms in the non-outliers part of the mode *i.e.,* equation Eq. (1). These are: the non-linear part of the mean $h_j(x)$ and the conditional variance $\sigma_j^2$. Mathematically, we simulate from the PPD of the response at location $s$:

$$p(\tilde{y}_{s,t}|\boldsymbol{y}) = \int_{\sigma_j^2, \boldsymbol{\gamma}_j, \boldsymbol{\phi}} p(\tilde{y}_{s,t}|\sigma_j^2, \boldsymbol{\gamma}_j, \boldsymbol{\phi}) p(\sigma_j^2|\beta_\sigma) p(\boldsymbol{\gamma}_j|\lambda_\gamma) p(\boldsymbol{\phi}|\boldsymbol{y}) d\sigma_j^2 d\boldsymbol{\gamma}_j d\boldsymbol{\phi} \tag{36}$$

**Table 2  Comparison of leave-one-out performance of the mean, 5% and 95% quantile in each station.** "Pred" relates to model estimates while "Obs" refers to the corresponding statistics of the observations. "Diff" is the difference between Obs and Pred.

| Station number | Mean | | | 2.5% quantile | | | 97.5% quantile | | |
|---|---|---|---|---|---|---|---|---|---|
| | Obs | Pred | Diff | Obs | Pred | Diff | Obs | Pred | Diff |
| 1 | 25.50 | 23.30 | 2.20 | 13.30 | 12.20 | 1.10 | 38.00 | 34.00 | 4.00 |
| 2 | 23.40 | 23.50 | −0.10 | 13.60 | 12.40 | 1.20 | 33.20 | 34.40 | −1.20 |
| 3 | 24.80 | 23.90 | 0.90 | 13.40 | 12.40 | 1.00 | 37.50 | 35.00 | 2.50 |
| 4 | 23.40 | 21.80 | 1.60 | 12.80 | 11.50 | 1.30 | 33.60 | 31.90 | 1.70 |
| 5 | 24.70 | 26.80 | −2.10 | 14.00 | 13.80 | 0.20 | 34.80 | 39.60 | −4.80 |
| 6 | 27.20 | 24.80 | 2.40 | 12.90 | 13.00 | −0.10 | 40.10 | 36.50 | 3.60 |
| 7 | 25.30 | 23.90 | 1.40 | 13.20 | 12.60 | 0.60 | 36.20 | 34.90 | 1.30 |
| 8 | 25.10 | 24.80 | 0.30 | 14.10 | 12.90 | 1.20 | 35.20 | 36.50 | −1.30 |
| 9 | 23.20 | 21.90 | 1.30 | 11.70 | 11.90 | −0.20 | 33.70 | 31.70 | 2.00 |
| 10 | 24.60 | 23.20 | 1.40 | 12.80 | 12.30 | 0.50 | 36.00 | 34.00 | 2.00 |
| 11 | 23.80 | 23.00 | 0.80 | 14.40 | 11.70 | 2.70 | 32.20 | 34.10 | −1.90 |
| 12 | 24.00 | 24.10 | −0.10 | 14.00 | 12.90 | 1.10 | 33.90 | 35.00 | −1.10 |
| 13 | 25.00 | 26.10 | −1.10 | 11.10 | 13.50 | −2.40 | 37.80 | 38.40 | −0.60 |
| 14 | 25.50 | 26.30 | −0.80 | 11.20 | 13.70 | −2.50 | 38.80 | 38.60 | 0.20 |
| 15 | 26.70 | 25.20 | 1.50 | 13.00 | 13.20 | −0.20 | 39.60 | 37.00 | 2.60 |
| 16 | 25.10 | 26.60 | −1.50 | 14.00 | 13.70 | 0.30 | 35.20 | 39.10 | −3.90 |
| 17 | 25.40 | 24.20 | 1.20 | 13.00 | 12.70 | 0.30 | 36.10 | 35.40 | 0.70 |

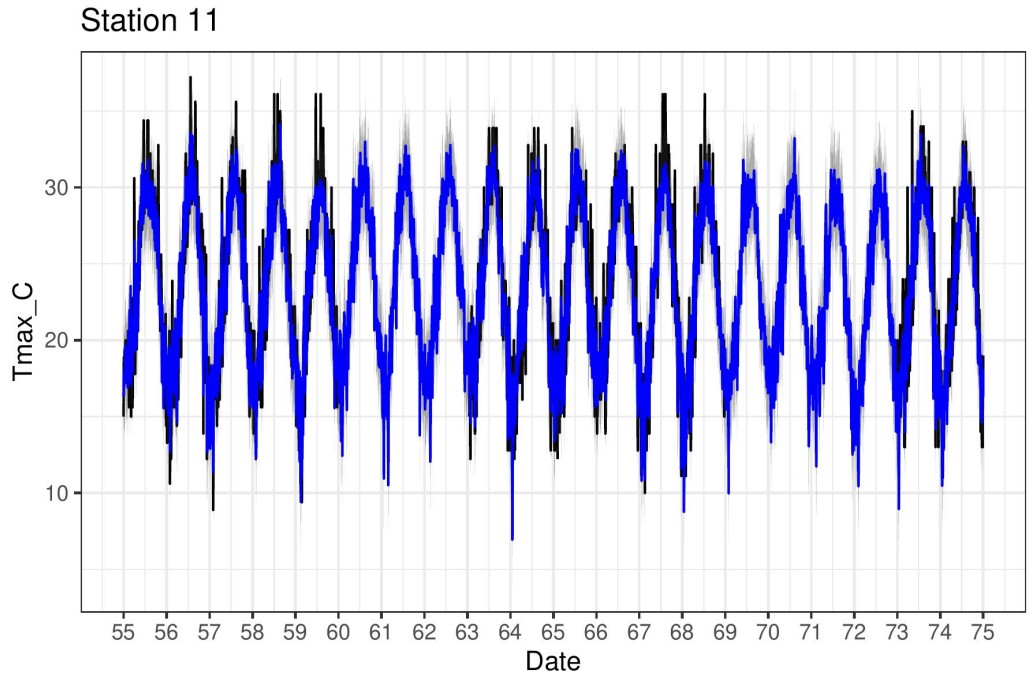

**Figure 7  Tmax time series (black) for station 11 (Pafos), with corresponding model predictions (blue).**

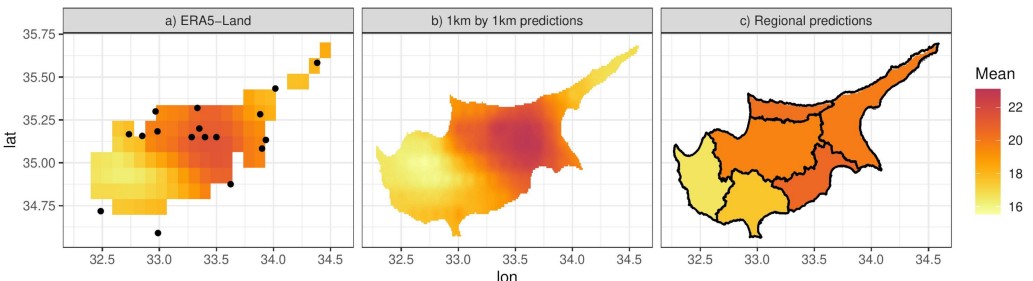

**Figure 8** (A) ERA-Land Tmax values for the 11th of April 2013. (B) Model predictions for the same day at 0.01° × 0.01° resolution. (C) Approximation of the integral of the predictions in (B) on the six districts of Cyprus.

where $\phi$ denotes all parameters other than $\sigma_j^2$ and $\gamma_j$. Since $h_j(x)$ is a function of $\gamma_j$, to integrate it out we need to simulate new $\gamma_j$'s from Eq. (18), where we use the R function `jagam` to set up the associated penalty matrix $S_\gamma$. Similarly we simulate "new" variances $\sigma_j^2$ for each unseen location using Eq. (25).

We illustrate this by predicting from the model at a grid of 0.01° × 0.01° (approximately 1 km × 1 km) resolution. Figure 8B shows the posterior predictive mean of the predictions for a particular day. Comparison with Fig. 8A, which shows the original ERA5-Land values for the same day, illustrates how the model can be used to downscale the reanalysis data, noting that the predictions from the model are also bias corrected (in a linear way). The non-linearity and station-specific variance have been integrated out and now are part of the prediction uncertainty. This uncertainty is quantified as the PPD standard deviation, shown in Fig. 9A. The contributing factors to the magnitude of this uncertainty are: station sparsity (*i.e.,* more uncertainty when predicting in areas far from weather stations), the conditional variance $\sigma_j^2$, and the integration of the non-linear part of the relationship with ERA5-Land. Figure 9B shows the standard deviation of the mean temperature Tmax *i.e.,* Eq. (4), which mirrors Fig. 9A indicating that the uncertainty due to the conditional variance is relatively small and spatially uniform. To understand the effect of non-linearity on the uncertainty, Fig. 9C shows the standard error of the mean, but without the non-linear term. This is clearly higher in regions with little data and *vice versa*. However, this uncertainty is small compared to when the non-linear terms is included, since the non-linear part adds uncertainty in regions where where ERA5-Land values are more extreme *e.g.,* the central region with the highest temperature values (see Fig. 8B) that nonetheless contains four stations. This indicates that the non-linear term $h_j(x)$ constitutes most of the prediction uncertainty.

Since $f(\cdot)$, $g(\cdot)$ and $h_j(x)$ are functions of random coefficients ($\alpha$, $\beta$ and $\gamma$), we can interpret the mean Eq. (4) and thus the predictions of the response as a random field. We can then integrate this random field over spatial regions, such as the six districts that make up the island of Cyprus (Fig. 8C). We can approximate this integral by simulating predictions at a high resolution grid (such as 0.01° × 0.01°) and then computing the mean of the simulations in each spatial unit. The higher the resolution, the better the approximation

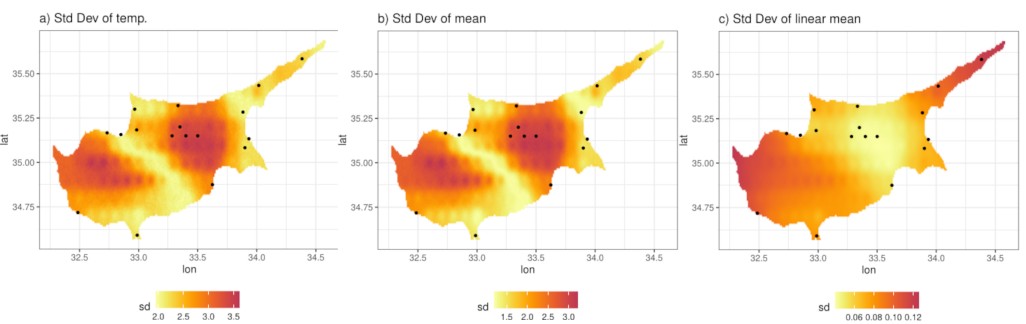

**Figure 9** (A) Standard deviation of the posterior predictive distribution for the 11th of April 2013. (B) Standard deviation of the mean. (C) Standard deviation of the mean, excluding the non-linear term.

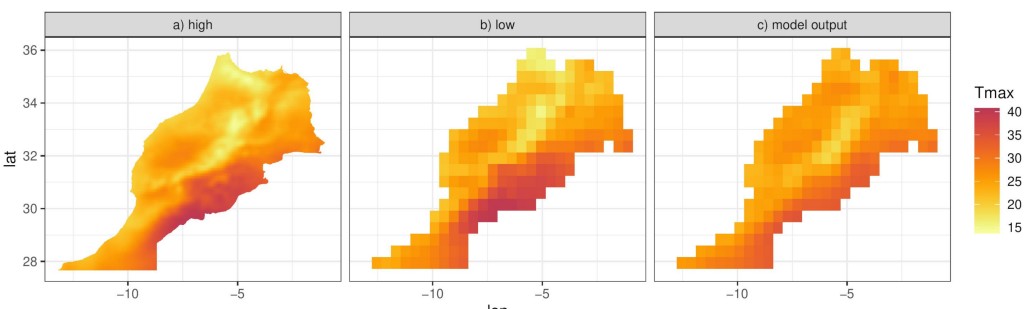

**Figure 10** (A) Model predictions for the 26th March 2005 at 0.01° × 0.01° resolution for Morocco. (B) Approximation of the integral of the predictions in (A), on a 0.44° × 0.44° resolution over Morocco. (C) Regional climate model (WRF) output of Tmax for the same day.

although for Cyprus we found virtually no difference in the results for resolutions higher than 0.01° × 0.01°. Figure 8C shows this approximation for the specific day, illustrating the ability of the model to predict at any spatial configuration. This also includes the ability to "upscale" the reanalysis to coarser resolutions for evaluating climate models. Figure 10 shows this for Morocco, where model predictions on a 0.01° × 0.01° grid are upscaled to 0.44° × 0.44° grid, the same grid corresponding to the output of a regional climate model (RCM) simulation. The RCM used to perform this simulation is the Weather Research and Forecasting (WRF) model (*Skamarock & Klemp, 2008*) driven by ERA-Interim reanalysis with a horizontal resolution of 0.44° (≈ 50 km) and 30 vertical levels, which was also used and evaluated in (*Constantinidou et al., 2020*). The model output is shown in Fig. 10C.

## SUMMARY AND DISCUSSION

We presented a probabilistic modelling framework to address certain requirements and challenges related to the use of temperature data from weather stations. The approach integrates climate model reanalysis data with *in situ* observations in a regression setting that allows for non-linearity in the relationship between the two. A discrete mixture formulation is used to identify non-physical outliers in the temperature observations so that associated

estimates can be used to "clean" the original data. It was demonstrated that the model can be used to impute missing values and to also produce predictions at any spatial location. The hierarchical nature of the framework allows for integration of station-specific effects in the predictions and this was used to produce a high resolution temperature map over Cyprus but also to integrate temperature measurements in contiguous spatial units.

The modelling framework was demonstratively flexible with very good in-sample and out-of-sample performance. The model could potentially be further improved, for instance to better capture particular aspects of the data such as extremes. One way might be to increase the number of components in the discrete mixture, and have one for extremes and one for non-extremes. Care needs to be taken however when the goal is to extrapolate. Complex modelling structures are more difficult to extrapolate to unseen locations in the covariate space, and is generally harder to constrain counter-intuitive behaviour. For instance, initial attempts here included the spatial extrapolation of both the linear and non-linear parts of the model and as a result predictions were in some cases nonsensical. Imposing the constraint that only the linear part is extrapolated avoided this issue. Note also how the uncertainty diagnosis in the results section indicated that most of the predictive uncertainty came from the non-linear part of the model, implying that model complexity translates to predictive out-of-sample uncertainty.

The change-of-support problem (*i.e.,* integrating point location data with gridded data) was dealt with by constructing a 10-neighbour weighted average of ERA5-Land for each station. The choice is subjective, and some sensitivity analysis is required for this choice. We found that for both Morocco and Cyprus, increasing the number of neighbours improves predictions and provides more smooth looking spatial structure when downscaling. However the improvement quickly plateaus and we found 10 to be an optimal choice. Ideally however, this choice can be dealt with in the model and future research is aimed at achieving this.

For a given station, the model uses both the information at stations and the reanalysis data to identify outliers and so can be used as an approach with which one can homogenise temperature records as well as impute missing values. By definition, it is impossible to really assess the ability of the model to identify the erroneous outliers, since these are truly unknown. Here we used intuition and physical understanding to judge this aspect of the model, in addition to a basic simulation experiment.

The ability to produce temperature estimates at any spatial resolution as a function of climate model output is an important aspect of the model. This opens up the possibility of addressing climate change by utilising historical data and model projections of future scenarios. This was not done as part of this article, which mainly concentrated at illustrating the framework, which in summary enables outlier detection, bias correction, downscaling and interpolation of temperature data. The unique ability of the approach to perform all these steps simultaneously in conjunction with quantifying uncertainty in a Bayesian manner, offers robust predictions that can be thoroughly evaluated. Future plans also include extension to "non-Gaussian" data such as precipitation and wind speed. Although this may take away much of the desired conditional conjugacy, it may be promising to consider Gaussian mixtures to preserve this while gaining non-symmetry.

# ACKNOWLEDGEMENTS

Neither the European Commission nor ECMWF is responsible for any use that may be made of the Copernicus information or data it contains.

### Funding

All authors were funded by the European Union's Horizon 2020 research and innovation programme under grant agreement No. 856612 and the Cyprus Government. The funders had no role in study design, data collection and analysis, decision to publish, or preparation of the manuscript.

### Grant Disclosures

The following grant information was disclosed by the authors:
European Union's Horizon 2020 research and innovation programme: 856612.
Cyprus Government.

### Competing Interests

The authors declare there are no competing interests.

### Author Contributions

- Theo Economou conceived and designed the experiments, performed the experiments, analyzed the data, prepared figures and/or tables, authored or reviewed drafts of the article, and approved the final draft.
- Georgia Lazoglou conceived and designed the experiments, performed the experiments, analyzed the data, prepared figures and/or tables, and approved the final draft.
- Anna Tzyrkalli conceived and designed the experiments, analyzed the data, prepared figures and/or tables, and approved the final draft.
- Katiana Constantinidou conceived and designed the experiments, prepared figures and/or tables, and approved the final draft.
- Jos Lelieveld conceived and designed the experiments, authored or reviewed drafts of the article, and approved the final draft.

### Data Availability

The data, code and supplementary material are available at Zenodo: Theo. (2022). Data, code and supplementary material for "A data integration framework for spatial interpolation of temperature observations using climate model data" (Version v2) [Data set]. Zenodo. https://doi.org/10.5281/zenodo.7294366.

The ERA5-Land data set (Munẽoz Sabater, 2021) is available from the Copernicus Climate Change Service (C3S) Climate Data Store. The results contain modified Copernicus Climate Change Service information 2020.

## Supplemental Information

Supplemental information for this article can be found online at http://dx.doi.org/10.7717/peerj.14519#supplemental-information.

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
