# Peer review of "A data integration framework for spatial interpolation of temperature observations using climate model data"

_PeerJ, doi:10.7717/peerj.14519_

## Round 0.1 · original submission · Minor Revisions

Dear Authors,

The reviewers have now commented on your manuscript and are suggesting certain improvements for making your manuscript more meaningful and better as far as the overall communication aspect of it is concerned.

Therefore, I suggest you kindly go through the reviewers' comments thoroughly and revise your manuscript accordingly.

I shall be looking forward to receiving your manuscript on time.

Best regards
Gowhar Meraj

Reviewer 1 ·

Basic reporting

1. Clear and crisp usage of English throughout the text. I congratulate the authors for producing this manuscript.
2. The introduction and the body of the text contain thorough references to previous research, which makes it easy to understand connection to related work.
3. The data sources are well-documented, and the authors have performed EDA on the weather stations in detail (like elevation, geographical distribution, NaN values, etc.), which helps ensure that the authors have tried to minimize bias in their data. This is especially relevant because their weather stations are scattered over a single country which has limited weather fluctuations.
4. The results are presented with clear labeling and lead to no ambiguity or confusion.
5. There is ample scope for future work, and the authors mention new applications and extensions of their work as well.

Experimental design

1. The impact of the research is clearly stated: to integrate temperature measurements by weather stations and climate data, to enable weather prediction at arbitrary spatial resolutions.
2. The addressed challenges are described in detail to allow for the full appreciation of the problems, and that of the resolution provided nby the current manuscript.
3. The probabilistic temperature model and the model for detecting outliers is chosen with a lot of deliberation and make use of previous work by several previous authors.
4. Model testing is done carefully with separate in-sample and out-of-sample performance measurements (I would have liked “K-fold” to be mentioned around like 314 to be clearer).
5. It is great that the authors have made a reference to using their method for detecting and predicting the effects of climate change. I had the thought throughout reading the manuscript, and I feel other readers would feel the same.

Validity of the findings

1. The probabilistic model they have considered is sound and clear reasoning is provided for their choices, with relevant references added in.
2. The authors provide a lot of enlightening discussions on handling missing data, outlier detection and statistical uncertainties.
3. The performance of their model against data is tested in various ways, and clear evidence is provided. It will be interesting to see performance of the present technique on global weather data as well, especially where the temperature ranges and fluctuations are much higher.
4. I congratulate the authors for this well-written manuscript, and recommend it for publication.

Additional comments

None

Reviewer 2 ·

Basic reporting

The manuscript is very well written with a clear narrative.

The literature review does a good job of placing the work within different parts the existing literature but could be clearer on how the proposed method compares to competitor approaches for solving this problem or indeed if they exist at all.

Line 66-71: I appreciate that you clearly set out your aims for the modelling approach, but could you briefly explain why each one is important for the application e.g. fully quantifying the uncertainty.

It would be interesting to know how long the model takes to run, how many data points there were, how many parameters were sampled, and was the MCMC coded by hand in R or did you use an MCMC software package.

Section references are missing numbers throughout the paper and there is inconsistent capitalisation.

Line 53: ‘Lack information about temperature at particular locations.' Could you be more explicit?

Experimental design

The model is elegant and the model checking is thorough. I have a couple of suggestions for the model design and a question about implementation.

Have you considered using tensor product smooths for f and g so that the smoothness penalty is different for longitude and latitude?

Is there scope to include any covariates in the spatial intercept or spatial slope of x for better prediction at unseen locations – altitude could make sense?

Does the choice of upper and lower bounds of the outlier generating function substantially impact the probability of individual data points being considered outliers?

Validity of the findings

Line 394: “it is impossible to really assess the ability of the model to identify 395 the erroneous outliers, since these are truly unknown.” Could you do a simulation experiment (e.g. time series only) to check the true and false positive rate etc, as well as the impact of the upper and lower bounds of the outlier distribution?

Additional comments

The R code contains pathways relating to one of the Author’s Dropbox.

There were some typos, see below.

Line 65: Typo ‘resolution.Specifically’

Line 77: ‘in a way to allow’ -> ‘in a way that allows’

Figure 2 appears to be cited after Figure 3 and Figure 4.

Line 241 typo: hyperameter

Line 285: ‘consider’ -> ‘considering’

Reviewer 3 ·

Basic reporting

The authors present an interesting problem statement and provide relevant motivation for solving this issue. A framework has been presented to integrate climate reanalysis and temperature data, in a probabilistic manner. The proposed Bayesian model quantifies uncertainty, is flexible in terms of capturing biases, helps in identifying outliers, in generating missing input and enables prediction at various spatial resolutions. The approach was implemented and studied on climate data from two regions- Middle East and North Africa region, which are major representative areas for drastic climatic changes.
The authors raise important points about the unavailability of temperature data at desired locations or for specific spatial resolutions, and how even if these data are available, they are prone to errors related to non-physical outliers, human errors introduced while manually processing historical data, etc.

The paper is well structured and easy to comprehend. The methodology, training strategies and results are well explained. Supplementary material includes models and additional data.

Suggested changes - In the ‘Introduction’ section, there exists a few typos in the last paragraph, where the section numbers are missing in the sentences although what the different sections contain are listed. For instance: “Section lays out the modeling framework,..”. In page 16/19, the figures could be brought to the same scale/dimensions.

Experimental design

The framework modelling details has clearly specified. Framework implementation involving sampling of outliers, outlier proportions, conditional variance, spline coefficients are clearly laid out. The experiments presented were robust. Exhaustive results from experiments have been provided which substantiates the objectives of this work. It is interesting to see the model was able to populate missing data for a period of time across stations, while also quantifying the uncertainty. It is also interesting to see that the prediction is great, even for isolated stations.
The supplementary data provided are replicable.

Validity of the findings

Drawbacks and challenges of using existing solutions such as ones presented by Durai and Bhradwaj 2014, Lazoglou et al., 2019, etc. have been discussed. Comparison results observed are satisfactory and present several key findings which proves the soundness of this framework. Limitations of this work have also been discussed such as issues in case of extrapolation and capturing trends at extremes.
Overall, the contributions of this paper are satisfactory, and the scientific community could greatly benefit from this framework.

---

## Round 0.2 · accepted · Accept

Dear Authors,

Thank you very much for incorporating all the suggestions commented on by the reviewers. I congratulate you on carrying out this rigorous work.

Thanks and best regards
Gowhar Meraj